# Impact of bictegravir/emtricitabine/tenofovir alafenamide on health-related quality of life and economic outcomes in HIV care: Substudy of the BIC-NOW clinical trial

Sergio Sequera-Arquelladas[1]*, María J. Vivancos[2], David Vinuesa[3], Antonio Collado[4], Ignacio De Los Santos[5], Patricia Sorni[6], Noemi Cabello-Clotet[7], Marta Montero[8], Carlos Ramos Font[9], Alberto Terron[10], Maria José Galindo[11], Onofre Martinez[12], Pablo Ryan[13], Mohamed Omar-Mohamed[14], Helena Albendín-Iglesias[15], Rosario Javier[1], Alberto Romero[16], Coral García Vallecillos[1], Miguel Ángel Calleja[17], Carmen Hidalgo-Tenorio[1]*

**1** Unit of Infectious Diseases, Hospital Universitario Virgen de las Nievesl, IBS-Granada, UGR, Granada, Spain, **2** Infectious Diseases Service, Hospital Ramón y Cajal, CIBERINFEC ISCIII, Madrid, Spain, **3** Unit of Infectious Diseases, Hospital Universitario San Cecilio, IBS-Granada, CIBERINFEC ISCIII, Granada, Spain, **4** Unit of Infectious Diseases, Hospital UniversitarioTorrecardenas, Almería, Spain, **5** Infectious Diseases Service, Hospital Universitario La Princesa, CIBERINFEC ISCIII, Madrid, Spain, **6** Unit of Infectious Diseases, Hospital Universitario Son Llàtzer, Palma de Mallorca, Palma, Spain, **7** Infectious Diseases Unit, Hospital clínico San Carlos, Complutense University, CIBERINFEC ISCIII, Madrid, Spain, **8** Infectious Diseases Service, Hospital Universitario La Fe, Valencia, Spain, **9** Nuclear Medicine Service, Hospital Universitario Virgen de las Nieves, Granada, IBS-Granada, Granada, Spain, **10** Unit of Infectious Diseases, Hospital Universitario Jerez, Cádiz, Spain, **11** Infectious Diseases Service, Hospital Universitario clínico Valencia, Valencia, Spain, **12** Unit of Infectious Diseases, Hospital Universitario Santa Lucía, Cartagena, Spain, **13** Internal Medicine Service, Hospital Universitario Infanta Leonor, CIBERINFEC, ISCIII, Madrid, Spain, **14** Complejo hospitalario Jaén, Jaén, Spain, **15** HIV and STI Unit, Department of Internal Medicine, Hospital Universitario Virgen de la Arrixaca, IMIB, Murcia, Spain, **16** Unit of Infectious Diseases, facultad de medicina, Hospital Universitario Puerto Real, INIBICA, Cadiz University, Cadiz, Spain, **17** Pharmacy in Hospital Universitario Virgen Macarena, Seville, Spain

* chidalgo72@gmail.com (CH-T); Sergio.Sequera@juntadeandalucia.es (SS-A)

## Abstract

### Background

The BICNOW clinical trial evaluated the effectiveness, safety, satisfaction, adherence to treatment, and retention in the system of a rapid initiation strategy with bictegravir/emtricitabine/tenofovir alafenamide (BIC/FTC/TAF) in naïve HIV-infected individuals. It also assessed the burden of this infection on individuals and healthcare systems using various instruments, participant questionnaires, and pharmacoeconomic evaluations of this antiretroviral therapy (ART). This substudy focused on changes in the health-related quality of life (HRQoL) of participants and on the economic impact of this rapid initiation strategy.

**Data availability statement:** All relevant data are within the manuscript and its Supporting Information files.

**Funding:** The grant was received by the public foundation associated with the coordinating hospital (FIBAO). Grant funded by Gilead Sciences Project ISR-ES-10-10727 https://www.gilead.es/ Funders did not intervene nor play any role in the study design, data collection and analysis, decision to publish, or preparation of the manuscript.

**Competing interests:** CHT has received honararia for consulting or speaking at educational events from Janssen, Gilead, MSD and ViiV.

## Methods and Findings

Patients were recruited for this phase IV, multicenter, open, single-branch clinical trial with 48-week follow-up between November 2020 and July 2022. HRQoL data were gathered using EQ-5D-3L and dichotomized HIV-SI questionnaires. In the cost-utility pharmacoeconomic analysis, data in the literature were used for comparators. The 208 participants had a mean age of 34 (27–44) years, 87·5% were male, 42·9% had completed higher education, and 67·1% were employed. The mean EQ-5D questionnaire score was significantly increased at 48 weeks versus baseline (0·940 ± 0·117 vs. 0·959 ± 0·083, p = 0·012), and the utility value in quality-adjusted life years (QALYs) was 0·877 ± 0·093. There was a significant improvement in the "usual activities" dimension (10·8 vs 4·1% p = 0·036). The Moses extreme reaction test showed a significant difference in all dimensions between participants in AIDS versus non-AIDS stage (p < 0·001). HIV-SI results revealed a significantly smaller percentage of participants with bothersome symptoms at 48 weeks (75·4 vs. 62·2%, p = 0·035). The pharmacoeconomic study indicated a value of €6,550·21/QALY gained with this ART.

## Conclusions

BIC/FTC/TAF is an appropriate rapid initiation strategy in naïve PLHIV that improves their quality of life. It is pharmacoeconomically feasible and offers superior long-term health outcomes in comparison to other approaches. (NCT06177574).

## Background

The burden of human immunodeficiency virus (HIV) infection remains a significant public health concern. It has multiple deleterious effects on society and healthcare systems, including a negative impact on the wellbeing of people living with HIV (PLHIV) [1].

Health-related quality of life (HRQoL) is a multidimensional concept that includes subjective evaluations of aspects of life and health that can exert a positive or negative influence on the quality of life. HRQoL is a useful instrument to evaluate chronic diseases, including HIV infection, and has been proposed as the "fourth 90" target in UNAIDS objectives [2].

Treatment strategies for PLHIV are under continuous development, and antiretroviral therapy (ART) has been universally recommended, regardless of the CD4 count or plasmatic viral load, since publication of the START trial results in 2015 [3]. Clinical guidelines even extend this recommendation to early initiation ARTswith specific drug combinations [4]. As a result, the life expectancy of PLHIV without comorbidities is virtually comparable to that of the general population [5], encouraging a focus on their quality of life. Thus, validated questionnaires are already available to evaluate HRQoL, including the generic EQ-5D questionnaire [6] and the PLHIV-specific HIV symptom index (HIV-SI) [7]. EQ-5D is a validated instrument to measure HRQoL and covers five dimensions: mobility, self-care, usual activities, pain/discomfort, and

anxiety/depression. Its simplicity and comprehensive nature facilitate the evaluation of treatment results from the perspective of participants, providing a direct view of the health and wellbeing of the individual [6]. HIV-SI is a validated instrument to gather the specific experience of symptoms by PLHIV. It considers a wide variety of HIV-associated symptoms and their treatment, allowing detailed analysis of their burden and time course under ART [7].

Pharmacoeconomic evaluation offers a comparative analysis of costs and consequences of alternative courses of action. Given the chronic nature of HIV and the need for lifelong treatment, ART evaluation is complex, addressing aspects that range from prevention to therapeutic options and self-care [8]. Calculations of the incremental cost-effectiveness ratio (ICER) and quality-adjusted life years (QALYs) provide highly relevant information for decision-making on investment in treatment strategies and offer precise feedback data on resource assignment decisions [8].

The objective of this substudy of the BICNOW clinical trial [9] was to evaluate HRQoL and pharmacoeconomic outcomes in naïve PLHIV started on a rapid initiation strategy with bictegravir/emtricitabine/tenofovir alafenamide BIC/FTC/TAF. The aim was to provide a general assessment of the HRQoL and economic implications of current HIV treatment in Spain.

## Methodology

### Design

This is a phase IV, open, single-branch, non-randomized, and multicenter substudy of the recently published BICNOW trial with48-week follow-up (NCT06177574) in naïve HIV individuals prescribed BIC/TAF/FTC as a test-and-treat strategy [9]. Patients were recruited between 16/11/2020 and 31/07/2022 from Departments/Units of Infectious Disease or Internal Medicine from 15 hospitals in the Spanish national health system.

### Interventions, variables, and origin of data

Participants completed EQ-5D-3L and HIV-SI questionnaires in their first post-recruitment visit and in follow-up visits at 24 and 48 weeks. S1 File.

The EQ-5D is a validated, widely used and cognitively undemanding generic health status questionnaire. It comprises two sections. The first (descriptive system) evaluates health in five dimensions (mobility, self-care, usual activities, pain/discomfort, and anxiety/depression) using a three-point response scale (1 = no problems, 2 = some problems, 3 = extreme problems/inability). The EQ-5D yields data that can be used to generate a health status profile [6], and these profiles can be directly related to a validated index score based on social preferences for health status, considering the Spanish context in the present study [10]. These weights, or utilities, are commonly used to calculate QALYs in economic health analyses. Health status index scores can range between <0 (0 = death and negative value = worse than death) and 1 (perfect health). The second section comprises a visual analog scale (VAS) for perceived health, with a score ranging from 0 (worst imaginable health) to 100 (best imaginable health) [10]. Mean scores with standard deviation (SD) were obtained from participants in the substudy at each follow-up visit.

The HIV-SI is a validated instrument that measures the symptom burden of PLHIV based on 20 ART-related symptoms, evaluating their impact on daily life [7]. Symptoms were dichotomized as symptomatic/bothersome or asymptomatic/non-bothersome to facilitate data gathering and simplify the analysis. The questionnaire results were divided into three categories: changes in individual symptoms, changes in the total number of bothersome symptoms, and changes in the presence of any bothersome symptom.Results for individual symptoms are used to identify areas for improvement in those that are most prevalent among participants

The questionnaires used to evaluate quality of life (EQ-5D, with 5 dimensions plus 1 VAS; and HIV-SI, with 20items) represent two validated measures rather than numerous independent assessment criteria. Each dimension or item is part of an interrelated set that captures specific aspects of quality of life or symptom burden. We therefore did not apply formal multiple comparison adjustments (e.g., Bonferroni or FDR), because these might unnecessarily inflate type II error rates for these intrinsically related dimensions. We focus on the main result (total score and/or key item) of each questionnaire,

providing subdomain results to supply additional clinical information. All p-values should be interpreted in the context of these correlated measures and item-level analyses.

Questionnaires with any error in their completion (e.g., marking of multiple boxes for the same item) were deemed invalid and were not scored or considered in the analysis.

## Pharmacoeconomics

In our evaluation of the disease burden for PLHIV and the healthcare system, we calculated QALYs, disability-adjusted life years (DALYs), and life years saved(LYS) in order to express the pharmacoeconomic results in a comprehensive and comprehensible manner [8]. We also carried out cost-utility and ICER analyses, selecting DRV/c/FTC/TAF and DTG/3TC as comparators because of their frequent utilization for rapid ART initiation in naïve individuals. Adjusted QALYs obtained in the present substudy were compared with the values for DRV/c/FTC/TAF and DTG/3TC reported by Butler K et al. [11]. Clinicaldata obtained in the BICNOW [9], DIAMOND [12] and STAT [13] trials served as reference values in our effectiveness analysis. We applied the laboratory retail prices recorded by the coordinating hospital in March 2024.

Mortality data for calculating the LYSand years of life lost (YLL) were extracted from the national database of the National Statistics Institute (*Instituto Nacional de Estadistica*, INE) [14] and Spanish Ministry of Health [15]. Disability weights were obtained from the study by Salomon J et al. 2013 [16].

## Methodology to calculate HRQoL variables

HRQoL was evaluated using the following indices, which center on distinct dimensions of mortality, morbidity, and quality of life:

YLL: Estimated life expectancy of the general population minus the life expectancy of individuals with HIV/AIDS under treatment. This index quantifies premature mortality, i.e., the number of potential life years lost due to HIV/AIDS in comparison to a population free of this disease [17].

Years of life with disability (YLD): Life expectancy with treatment multiplied by the burden of disability for HIV/AIDS under treatment. This index indicates the years lived with reduced health due to HIV/AIDS, even under treatment [17].

DALYs (YLL+YLD): Total disease burden, combining premature mortality (YLL) and lifetime with disability (YLD). One DALY represents the loss of one year of "perfect" health [17].

LYS: Life expectancy with treatment minus life expectancy without treatment, indicating the additional years of life gained by treatment [18].

QALYs: This index integrates the quantity of life with the quality of life, with a higher score indicating longer and higher quality survival [18].

Discounted QALYs adjusted by total life years (LY) and LYS: This measurement offers a long-term perspective of the net benefits of intervention, considering an increase in both the life expectancy and the quality of additional years gained. The discount indicates the value of receiving health benefits sooner rather than later, assisting resource assignment and HIV treatment strategy optimization [18].

The proportional annual discount applied to costs and utilities is 3%, as recommended in the guidelines of the Institute for Clinical and Economic Review [19].

## Sample size

We performed a new calculation for this substudy. Previous reports indicated that SD = 0·4 can be expected in sample size estimations for HRQoL analysis. It was estimated that 208 participants would be sufficient to meet substudy objectives with 95% confidence interval and 5% error margin, considering losses to the study of up to 10%. The supplementary appendix in S1 File provides further details of the sample size calculation.

## Statistical analysis

In the descriptive analysis, we calculated means with SD for quantitative variables with normal distribution (Cramer-von Mises test) and medians with interquartile range (IQR) for those with non-normal distribution. Qualitative variables were expressed as absolute and relative frequencies. In the bivariate analysis, we used the chi-square test to compare qualitative variables with normal distribution, the Mann–Whitney U test for quantitative variables with non-normal distribution, and the Student's t-test for quantitative variables with normal distribution. The Moses extreme reaction test was used to explore between-group differences (AIDS vs. non-AIDS) [20].

## Ethics approval and consent to participate

This study was conducted in accordance with the Declaration of Helsinki and was approved by the Ethics Committee of Granada (CEImGranada) on 27 October 2020 (Acta 13/20). Written informed consent was obtained from all participants before inclusion

## Results

### Description of the substudy population

The substudy included 208 naïve PLHIV with a mean age of 34 years (27–44), 87·5% were male, 70·2% were men who have sex with men (MSM); 79·3% had secondary or higher education and 66·8% were employed; 22·6% were in AIDS stage (Table 1). A more detailed description of participants in the BICNOW trial can be consulted in https://doi.org/10·1016/j.ijantimicag.2024·107164 [9].Among the 208 participants, 178 (85·6%) completed the 48-week follow up with 169 EQ-5D and 172 HIV-SI questionnaires deemed valid (Fig 1).

### HRQoL results: EQ-5D score

EQ-5D questionnaire findings revealed a significantly decreased severity in the HRQoLdimension of usual activities at 48 weeks(10·8*vs.* 4·1%, MD=0.71, 95% CI [0·005, 0·137], p=0·036). No statistically significant differences were observed in mobility (5·4*vs.* 3%, p=0·248), self-care (2·2*vs.*1·2%, p=0·334), pain/discomfort (16·1*vs.* 11·8%, p=0·274), or depression (26·3*vs.* 22·5%, p=0·114). Although a higher percentage of participants scored their quality of life as "perfect health status"(profile 11111=no limitation in any dimension)at week 48, the difference was not statistically significant

**Table 1. Demographic characteristics.**

| Demographic characteristics | N=208 |
|---|---|
| Age (years), median (IQR) | 34 (27-44) |
| Males, n (%) | 182 (87·5) |
| *Maximum educational level reached, n (%)* | |
| Reading and writing | 1 (0·48) |
| Elementary/primary | 42 (20·19) |
| Secondary/vocational | 76 (36·54) |
| Higher/university | 89 (42·79) |
| *Employment status, n (%)* | |
| Employed/freelance | 139 (66·83) |
| Home care | 11 (5·29) |
| Student | 21 (10·10) |
| Unemployed | 31 (14·90) |
| Retired | 6 (2·88) |

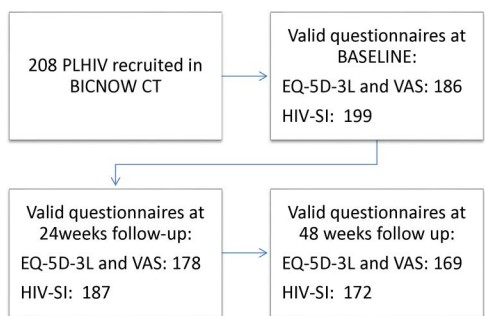

**Fig 1. Trend flow.** Valid questionnaires per visit.

(65·6*vs.* 69·2%, p = 0·194); however, this change reached significance among the non-AIDS participants (65·3*vs.* 70·87%, MD = −0·084, 95% CI [−0·185, 0·017], p = 0·031, 1-tailed)).

The mean utility calculated by index scores was significantly improved at 48 weeks (0·940 ± 0·117 *vs.* 0·959 ± 0·083, MD = −0·023, 95% CI [−0·045, −0·001] p = 0·012). The QALY calculated with this utility was 0·877 ± 0·093 QALY. Table 2 lists remaining variables.

No difference in self-perceived health was found between baseline and week 48 when all participants were considered (85·1 ± 14·0 *vs.* 86·8 ± 11·9, p = 0·148), but a statistically significant change was observed among the participants in non-AIDS stage (85 ± 1·24 vs 86·5 ± 1·1, MD = −1·67 95%CI [−2·97, 0·03], p = 0·046, 1-tailed) (Table 2). When scores were stratified as < 80, 80–89, or ≥ 90, a significant improvement was found in the number of individuals with a score < 80 at week 24 (22 *vs.* 14%, MD = 0·81, 95% CI [0·014, 0·148], p = 0·019) (Fig 2).

The Moses extreme reaction test results evidenced significant differences in EQ-5D and VAS scores between participants in AIDS *versus* non-AIDS stage (p < 0·001 in all), confirming a significant between-group variation in HRQoL (Supplementary appendix in S1 File).

Complete tables with mean differences and CIs are available in the supplementary appendix, S1 File.

At week 48, 9 EQ-5D-3L questionnaires were deemed invalid.

**Table 2. EQ-5D-3L – results per evaluated dimension.**

|  | BASELINE N = 186 | 24W N = 178 | 48W N = 169 | p-value (BASELINE vs 48W) N = 169 |
|---|---|---|---|---|
| **Mobility, n (%)** | 10 (5·38) | 3 (1·69) | 5 (2·96) | 0·248 |
| **Self-care, n (%)** | 4 (2·15) | 2 (1·12) | 2 (1·18) | 0·334 |
| **Usual activities, n (%)** | 20 (10·75) | 12 (6·74) | 7 (4·14) | **0·036** |
| **Pain/discomfort, n (%)** | 30 (16·13) | 28 (15·73) | 20 (11·83) | 0·274 |
| **Anxiety/depression, n (%)** | 49 (26·34) | 53 (29·78) | 38 (22·49) | 0·114 |
| **Perfect heath status 11111, n (%)** | 122 (65·59) | 110 (61·80) | 117 (69·23) | 0·194 |
| **AIDS** | 25 (65·79) | 29 (67·44) | 26 (63·41) | 0·5 |
| **NON-AIDS** | 97 (65·31) | 81 (60·45) | 91 (70·87) | **0·031*** |
| **Calculated utilities, mean (±SD)** | 0·94 (0·12) | 0·95 (0·08) | 0·959 (0·09) | **0·012** |
| **VAS score, mean (±SD)** | 85·08 (14·02) | 85·72 (13·92) | 86·80 (11·93) | 0·148 |
| **AIDS** | 85·25 (2·17) | 86·33 (1·88) | 86·37 (1·77) | 0·305 |
| **NON-AIDS** | 84·97 (1·24) | 86·90 (1·10) | 86·52 (1·09) | **0·046*** |

*: Exact Sig. (1-tailed); AIDS : acquired immune deficiency syndrome; VAS : Visual analog scale.

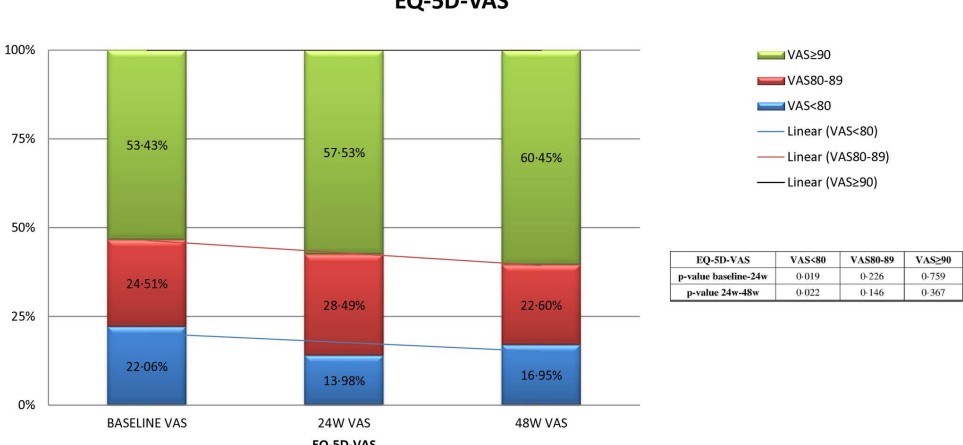

**Fig 2. EQ-5D-VAS Results.** With trend lines and p values.

## HRQoLresults: HIV-SI score

HIV-SI questionnaire results showed a significantly reduced percentage of participants with any bothersome symptom between baseline and 48 weeks (78·4 *vs.* 64·8%, MD = 0·131, 95% CI [0·052, 0·211], p = 0·001), and quantitative analysis revealed a reduction in the median number of bothersome symptoms (2 [1–5] *vs.* 1 [0–4, MD = 0·503, 95% CI [−0·06, 1·065]], p = 0·039, 1-tailed). The only symptoms showing a significant reduction at week 48 were diarrhea (17·1 *vs.* 8·5%, MD = 0·071, 95% CI [0·008, 0·135], p = 0·043), loss of appetite (11·06 vs 5·03%, MD = 0·658, 95% CI [0·124, 0·119], p = 0·027) and nervousness/anxiety (28·6 *vs.* 21·1%, MD = 0·071, 95% CI [−0·04, 0·147], p = 0·044, 1-tailed). Table 3 and Fig 3 lists remaining variables.

Complete tables with mean differences and CIs are available in the supplementary appendix, S1 File.

At week 48, 6 HIV-SI questionnaires were deemed invalid

## Pharmacoeconomic analysis

The estimated cost of utility for BIC/FTC/TAF treatment in our study was €6,550·21 per QALY gained or DALY averted.

The LYS due to the ART was 27·7 additional years, the potential YLL was 38·1 years, and the YLD valuewas5·7 years. This represents a total disease burden of 43·8 years (DALYs) considering both premature death and disability.

In the utility analysis adjusted by the additional years lived due to the treatment (LYS), the discounted QALY value was around 17; in the analysis adjusted for useful life years, it was around 20·6.

## Economic evaluation

Discounted QALY values were 20·6, 18·1, and 17·9 QALYs for BIC/FTC/TAF, DTG/3TC, and DRV/c/FTC/TAF, respectively. The effectiveness (per-protocol analysis) was 98·3% [9], 92% [13], and 96% [12] and the CER was €278·60, €268·56, and €340·08; respectively.

The utility-based ICER was similar between BIC/FTC/TAF and DTG/3TC, at€350·16 per lost QALY, and between BIC/FTC/TAF and DRV/c/FTC/TAF, at €-132·63€ per lost QALY (Table 4).

## Discussion

Most participants were young MSM in employment with a medium-high educational level. After starting BIC/FTC/TAF as a rapid initiation strategy, participants experienced a significant improvement in quality of life, and there was a lower

**Table 3. Results of dichotomized HIV-SI questionnaire (bothersome/non-bothersome).**

| | BASELINE N = 199 | W24 N = 187 | W48 N = 172 | p-value (BASELINE vs 48W) N = 172 |
|---|---|---|---|---|
| **Any bothersome symptom, n (%)** | 156 (78·39) | 150 (75·38) | 129 (64·82) | **0·001** |
| **Overall bothersome symptoms, median (IQR)** | 2 (1-5) | 2 (0-4) | 1 (0-4) | **0·039*** |
| **Fatigue/loss of energy, n (%)** | 47 (23·62) | 42 (21·11) | 30 (15·08) | 0·160 |
| **Fevers/chills/sweats, n (%)** | 20 (10·05) | 16 (8·04) | 16 (8·04) | 0·395 |
| **Dizzy/light headedness, n (%)** | 24 (12·06) | 18 (9·05) | 19 (9·55) | 0·565 |
| **Pain/numbness/tingling in hands/ feet, n (%)** | 26 (13·07) | 26 (13·07) | 28 (14·07) | 0·716 |
| **Difficulty remembering, n (%)** | 15 (7·54) | 18 (9·05) | 19 (9·55) | 0·373 |
| **Nausea/vomiting, n (%)** | 20 (10·05) | 14 (7·04) | 10 (5·03) | 0·226 |
| **Diarrhea/loose bowels, n (%)** | 34 (17·09) | 26 (13·07) | 17 (8·54) | **0·043** |
| **Sad/feeling down/depressed, n (%)** | 52 (26·13) | 39 (19·6) | 44 (22·11) | 0·249 |
| **Nervous/anxious, n (%)** | 57 (28·64) | 55 (27·64) | 42 (21·11) | **0·044*** |
| **Difficulty sleeping, n (%)** | 56 (28·14) | 56 (28·14) | 56 (28·14) | 0·764 |
| **Skin problems/rash/itching, n (%)** | 33 (16·58) | 34 (17·09) | 23 (11·56) | 0·158 |
| **Coughing/trouble breathing, n (%)** | 15 (7·54) | 18 (9·05) | 11 (5·53) | 0·319 |
| **Headaches, n (%)** | 39 (19·6) | 29 (14·57) | 34 (17·09) | 0·413 |
| **Loss of appetite, n (%)** | 22 (11·06) | 15 (7·54) | 10 (5·03) | **0·027** |
| **Bloating/pain/gas in stomach, n (%)** | 36 (18·09) | 34 (17·09) | 31 (15·58) | 0·623 |
| **Muscle aches/joint pain, n (%)** | 38 (19·1) | 37 (18·59) | 31 (15·58) | 0·614 |
| **Problems with sex, n (%)** | 37 (18·59) | 27 (13·57) | 25 (12·56) | 0·145 |
| **Changes in body composition, n (%)** | 39 (19·6) | 42 (21·11) | 39 (19·6) | 1 |
| **Weight loss/wasting, n (%)** | 17 (8·54) | 14 (7·04) | 13 (6·53) | 0·656 |
| **Hair loss/changes, n (%)** | 21 (10·55) | 29 (14·57) | 28 (14·07) | 0·355 |

*: Exact Sig. (1-tailed); IQR : Inter quartile range; HIV - SI : Human immunodeficiency virus symptom index

percentage of bothersome symptoms, with a marked reduction in the frequency of diarrhea and a recovery of appetite. Their quality of life became comparable to that described in the general populations of Spain [21] and Europe [22].

Between baseline and 48 weeks, increases were observed in the EQ-5D utility scores and VAS scores of participants, contributing further evidence of the importance of early ART initiation in individuals diagnosed with HIV, improving their quality of life and general health status [23].

Moses extreme reaction test results evidenced a significant variability in HRQoL scores, with low scores being more common among participants in AIDS stage. This finding illustrates the need to direct interventions towards the specific requirements of PLHIV at different stages of the disease. It has previously been reported that the AIDS stage has a considerable influence on the quality of life of PLHIV [24].

We highlight the significant reduction in the total percentage of bothersome symptoms gathered by the HIV-SI questionnaire and the significant or non-significant decrease in the percentage of patients individually reporting bothersome physical, cognitive, or psychological symptoms. This decline in the disease burden of patients can be attributed to the virologic and immunologic effects of the ART [25]. Nevertheless, some symptoms persisted in more than half of the participants at

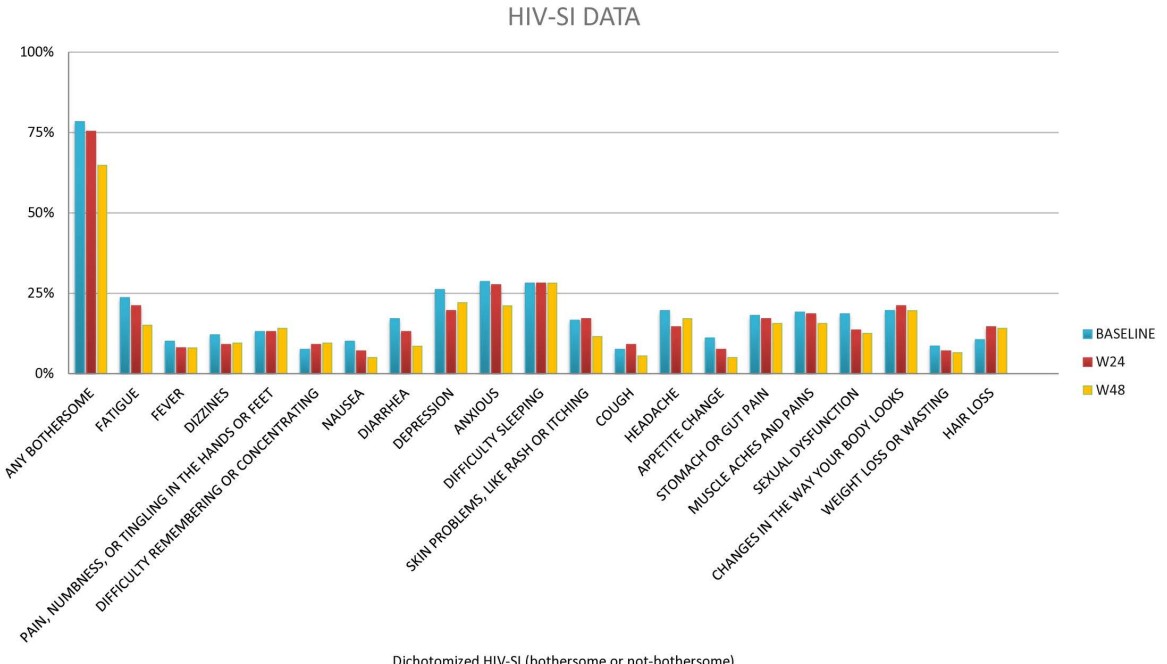

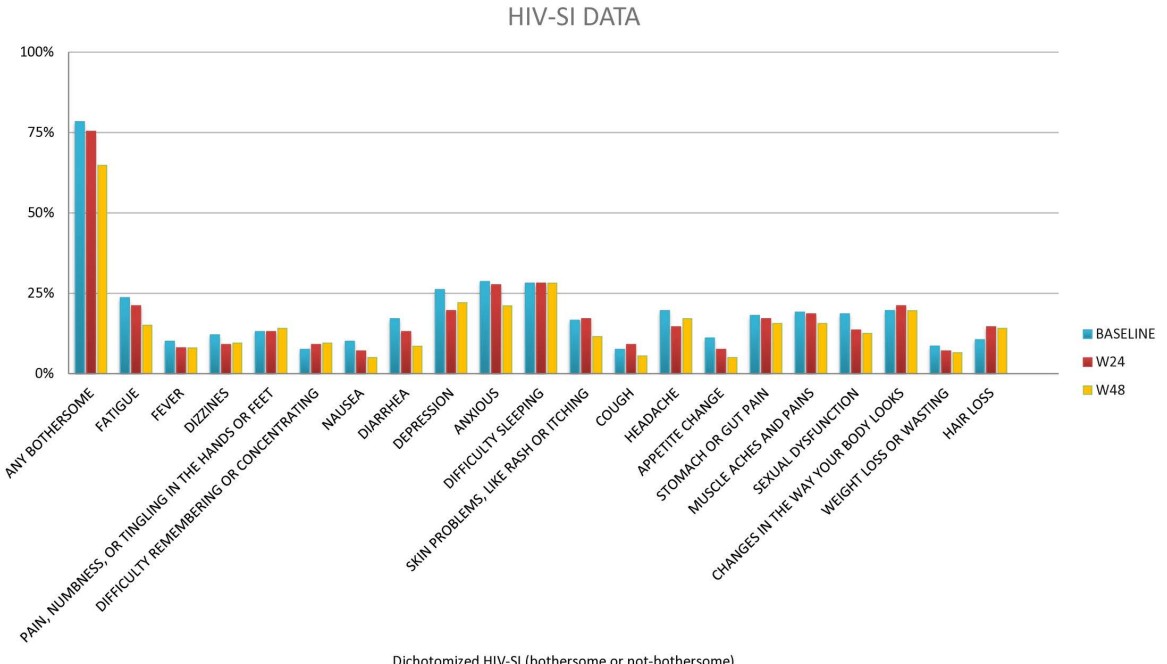

**Fig 3. Graph of HIV-SI Results.** Bars indicate percentage of bothersome outcomes.

**Table 4. Pharmacoeconomic analysis.**

| | QALY adjusted by LY | Effectiveness PP | CEA | CUA | ICER-u |
|---|---|---|---|---|---|
| DTG/3TC | 18·06 [11] | 92[13] | 52·73 | 268·56 | 350·16 |
| BIC/FTC/TAF | 20·60 | 98·3[9] | 58·38 | 278·60 | – |
| DRV/c/FTC/TAF | 17·92[11] | 96[12] | 63·48 | 340·08 | −132·63 |

QALY: quality adjusted life year; LY: total life year; PP: per-protocol analysis; CEA: Cost economic analysis; CUA: Cost Utility analysis; ICER-u: Incremental cost-effective ratio – utilities applied: DTG: Dolutegravir; 3TC: Lamivudine; BIC: Bictegravir; FTC: emtricitabine; TAF: Tenofovir alafenamide fumarate; DRV/c: Cobicistat boosted darunavir.

48 weeks, most frequently sleeping disorders, depression, and nervousness/anxiety. These were also the most commonly reported symptoms in other studies of naïve patients [3]. Likewise, the disease burden calculated from the adjustedYLL and DALY data is in agreement with previously published findings on the consequences of the disease in PLHIV [1,26]. These data confirm the considerable health burden of HIV infection described by the WHO, with HIV remaining as one of the 20 main causes of DALYs worldwide [27].

Therapid initiation strategy has proven to be an effective, safe and fast approach that reduces the viral load, promotes immunological reconstruction, and improves adherence to the healthcare system [9].Hence, a major priority now is to develop and implement strategies for ameliorating the quality of life of patients. In this substudy, we identified areas for improvement in physical (fatigue, pain) and cognitive (anxiety, sleeping disorders) dimensions, in line with previous reports [1]. Approaches to these problems include the application of concomitant therapies with demonstrated effectiveness, such as analgesic treatments against chronic pain, physiotherapeutic support for physical symptoms and psychological support for psychoaffectivesymptoms, as recommended in the literature and clinical care guidelines [28,29].

The pharmacoeconomic analysis in our substudy contributes valuable information on the economic and general impact of ART with BIC/FTC/TAF. The estimated utility cost for treatment in our analysis is €6,550·21 per QALY gained or DALY averted, significantly below the threshold of €30,000 per QALY established for Spain [30]. When the calculated utility (0·877 QALY) is adjusted to a 3% discount rate in total life years, the result is 20·6 QALYs, a slightly higher quality-adjusted life expectancy in comparison to the comparators considered in this study.

Although our pharmacoeconomic study indicates that therapy with DTG/3TC is the most cost-effective (lower CER and positive ICER), we need to consider the per-protocol effectiveness as analyzed in the reference studies. The high per-protocol effectiveness described for BIC/FTC/TAF in the BICNOW trial (almost 100%) indicates a better control of disease progression, but this combination is not the least costly in absolute terms. Overall, our results do not show a sufficiently significant difference between BIC/FTC/TAF and comparators to rule out alternative rapid treatment options, and each may be more appropriate in certain scenarios [31]. Among the three drug combinations considered, DTG/3TC offers the lowest cost per QALY, offering the greatest gain in utility per euro spent. The positive ICER value indicates that switching from BIC/FTC/TAF to DTG/3TC would reduce costs but at the expense of losing QALYs. Thus, the small saving in costs would not be sufficient to compensate for the worse long-term health outcomes obtained in comparison to BIC/FTC/TAF. The economic results appear to be least favorable for DRV/c/FTC/TAF (negative ICER), although this combination may be more appropriate in specific clinical situations.

This substudy is mainly limited by its open non-randomized design, although this limitation is shared with the other clinical trials on rapid initiation in naïve PLHIV (DIAMOND [12] and STAT [13]) Furthermore, the follow-up is restricted to 48 weeks, and longer-term research is needed. Its strengths include: the use of patient reported outcomes, which extends the disease burden assessment and adds depth to the analyses; the largest sample size published to date for a rapid initiation model of care in naïve PLHIV; and its combined analysis of HRQoL and pharmacoeconomic results, which has only rarely been performed. Finally, the utilization of EQ-5D and HIV-SI questionnaires promotes the integration of economic evaluation with person-centered care, underscoring the importance of both clinical effectiveness and the subjective wellbeing of PLHIV.

## Conclusions

BIC/FTC/TAF is an optimal treatment option for the rapid initiation model, improving the clinical outcomes and HRQoL of PLHIV and yielding the economic benefits associated with rapid initiation strategies. In comparison to DTG/3TC, BIC/FTC/TAF offers superior long-term health results in naïve PLHIV at a slightly higher cost. These findings confirm that the rapid diagnosis and treatment of naïve PLHIV is of critical importance to improve their health outcomes, functionality, and wellbeing.

## Supporting information

**S1 File. Supplementary appendices containing additional tables, figures, and methodological details referenced in the main manuscript.**
(ZIP)

**S1 Data. Completed TREND statement checklist documenting compliance with reporting guidelines for nonrandomized evaluations.**
(PDF)

**S2 Data. Detailed study protocol describing study design, interventions, outcomes, and statistical analyses.**
(PDF)

## Acknowledgments

The authors are grateful to the people living with HIV and their family members for their generous participation in the study, Medicine Department. University of Granada, Spain.

## Author contributions

**Conceptualization:** Carmen Hidalgo-Tenorio.

**Data curation:** María J. Vivancos, David Vinuesa, Antonio Collado, Ignacio De Los Santos, Patricia Sorni, Noemi Cabello-Clotet, Marta Montero, Carlos Ramos Font, Alberto Terron, Maria José Galindo, Onofre Martínez, Pablo Ryan, Mohamed Omar-Mohamed, Helena Albendín-Iglesias, Rosario Javier, Alberto Romero, Coral García Vallecillos.

**Formal analysis:** Coral García Vallecillos, Carmen Hidalgo-Tenorio.

**Investigation:** Sergio Sequera-Arquelladas, María J. Vivancos, David Vinuesa, Antonio Collado, Ignacio De Los Santos, Patricia Sorni, Noemi Cabello-Clotet, Marta Montero, Carlos Ramos Font, Alberto Terron, Maria José Galindo, Onofre Martínez, Pablo Ryan, Mohamed Omar-Mohamed, Helena Albendín-Iglesias, Rosario Javier, Alberto Romero, Carmen Hidalgo-Tenorio.

**Methodology:** Sergio Sequera-Arquelladas, Coral García Vallecillos, Carmen Hidalgo-Tenorio.

**Project administration:** Sergio Sequera-Arquelladas, Carmen Hidalgo-Tenorio.

**Resources:** María J. Vivancos, David Vinuesa, Antonio Collado, Ignacio De Los Santos, Patricia Sorni, Noemi Cabello-Clotet, Marta Montero, Carlos Ramos Font, Alberto Terron, Maria José Galindo, Onofre Martínez, Pablo Ryan, Mohamed Omar-Mohamed, Helena Albendín-Iglesias, Rosario Javier, Alberto Romero, Miguel Ángel Calleja.

**Software:** Coral García Vallecillos.

**Supervision:** Miguel Ángel Calleja.

**Validation:** Sergio Sequera-Arquelladas, María J. Vivancos, David Vinuesa, Antonio Collado, Ignacio De Los Santos, Patricia Sorni, Noemi Cabello-Clotet, Marta Montero, Carlos Ramos Font, Alberto Terron, Maria José Galindo, Onofre Martínez, Pablo Ryan, Mohamed Omar-Mohamed, Helena Albendín-Iglesias, Rosario Javier, Alberto Romero.

**Visualization:** Miguel Ángel Calleja.

**Writing – original draft:** Sergio Sequera-Arquelladas, Carmen Hidalgo-Tenorio.

**Writing – review & editing:** Sergio Sequera-Arquelladas, Carmen Hidalgo-Tenorio.

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
