## [Decision Letter · Decision Letter 0]

3 Jan 2025

Dear Dr. Sequera,

**You will see that the Referees found your work of some interest. However, they also raised major criticisms. Please respond to all the comments by Reviewers, with special attention to methodological points raised by Reviewer #1 and Reviewer #2.**

Please submit your revised manuscript by Feb 17 2025 11:59PM. If you will need more time than this to complete your revisions, please reply to this message or contact the journal office at plosone@plos.org. A rebuttal letter that responds to each point raised by the academic editor and reviewer(s). You should upload this letter as a separate file labeled 'Response to Reviewers'.A marked-up copy of your manuscript that highlights changes made to the original version. You should upload this as a separate file labeled 'Revised Manuscript with Track Changes'.An unmarked version of your revised paper without tracked changes. You should upload this as a separate file labeled 'Manuscript'.

We look forward to receiving your revised manuscript.

Kind regards,

Giuseppe Vittorio De Socio, MD, PhD

Academic Editor

PLOS ONE

**Journal Requirements:**

3. Please amend the manuscript submission data (via Edit Submission) to include authors Dr. María J. Vivancos, Dr. David Vinuesa, Dr. Antonio Collado, Dr. Ignacio De Los Santos, Dr. Patricia Sorni, Dr. Noemi Cabello-Clotet, Dr. Marta Montero, Dr. Carlos Ramos Font, Dr. Alberto Terron, Dr. MJ Galindo, Dr. Onofre Martinez, Dr. P.Ryan, Dr. Mohamed Omar-Mohamed, Dr. Helena Albendín-Iglesias, Dr. Rosario Javier, Dr. Miguel Ángel López- Ruz, Dr. Alberto Romero, Dr. Coral García Vallecillos, Dr. MA Calleja and Dr. Carmen Hidalgo-Tenorio.

6. Please include your tables as part of your main manuscript and remove the individual files. Please note that supplementary tables (should remain/ be uploaded) as separate "supporting information" files.

Reviewers' comments:

Reviewer's Responses to Questions

**Comments to the Author**

1. Is the manuscript technically sound, and do the data support the conclusions?

Reviewer #1: No

Reviewer #2: Yes

2. Has the statistical analysis been performed appropriately and rigorously?

Reviewer #1: No

Reviewer #2: Yes

3. Have the authors made all data underlying the findings in their manuscript fully available?

Reviewer #1: Yes

Reviewer #2: No

4. Is the manuscript presented in an intelligible fashion and written in standard English?

Reviewer #1: Yes

Reviewer #2: Yes

**Reviewer #1: ** General comments:

These analyses explore quality of life measures in a subset of clinical trial participants. Comparisons are made between participant scores 24- and 48-weeks post-recruitment.

I have some concerns with these analytic methods. First, there is no attempt to control for confounders in any of the analyses. All differences are attributed to time (24 vs. 48 weeks post recruitment) and no other factors that may have changed are considered. A multivariable model that controls for other factors would be beneficial.

A multivariable model would also help my second concern. I think the manuscript would benefit from having comparisons reported as differences with CIs. For instance, I found it hard to understand the difference in line 234 since, based on the SDs, it looks like the intervals might overlap. Reporting differences as mean differences or odds ratios with a CI would facilitate understanding how comparisons were made.

Third, why is baseline not included in analyses? I would have expected that comparisons would be made to baseline whenever possible, yet the analyses and write up suggest baseline does not factor in to these analyses. I would love to see longitudinal analyses performed, probably with random effects, where confounders are included in the models and you could estimate 24w v. BL, 48w v. BL, and 48w v. 24w.

Third, I found the reporting of missing data and the sample sizes for analyses confusing. For instance, lines 215-216 have 208 people at 24-weeks and 178 at 48-weeks. That indicates to me that the two time points have different sample sizes. Thus, the differences quoted in lines 225-227 could be explained by participant drop out and not necessarily differences between 24- and 48-weeks. It's even worse in Tables 2 and 3 where the sample sizes are never 208. Differences in the sample size may not contribute to differences, but as currently presented, the differing sample sizes raise doubt on your results. I strongly recommend using the same sample size and participants for all analyses in a subset.

In addition, if there are only 178 people, that means, based on your sample size, you are underpowered for your analyses. In addition, the sub-subgroup analyses of AIDS and non-AIDS people are further underpowered. Were there power analyses performed for these sub-subgroup analyses?

Lastly, there are LOTS of evaluations being done here. I would think some sort of p-value correction should be performed to account for multiple testing. For instance, something controlling for the false discovery rate would be useful.

Specific comments:

1. (line 55) "p<" needs to be changed.

2. (lines 136-138) Are the symptoms dichotomized in the survey instrument? If they are not dichotomized in the instrument, then I believe this would invalidate the validation that was done on the instrument since the instrument is not used as developed.

3. (lines 141-143) How many questionnaires were deemed invalid?

4. (line 195) Although the Kolmogorov–Smirnov test is frequently used, it is well-known to be underpowered for testing normality. Other tests have more power (e.g., see https://www.jstor.org/stable/2286009), such as Andersen-Darling and Cramer von Mises.

5. (lines 195-202) I would prefer providing methodological citations for each of the tests here to credit test developers for their work, though it seems to be ok to not cite commonly used tests. I am familiar with all tests in this section, except for the Moses extreme test, so I would suggest providing a methodological citation for that test.

6. (lines 219-227) I'm assuming the comparisons in this paragraph are between the 24- and 48-week outcomes, but it's not completely clear.

7. (line 224) What is "profile 11111"?

8. Why are the tables out of order?

9. (Figure 2) I don't think this figure adds anything that cannot be explained in the text.

**Reviewer #2: ** General Comments:

This manuscript addresses an important topic in HIV care by evaluating the impact of rapid initiation of bictegravir/emtricitabine/tenofovir alafenamide (BIC/FTC/TAF) on health-related quality of life (HRQoL) and economic outcomes. The findings contribute useful information regarding clinical and pharmacoeconomic aspects of this treatment strategy. The manuscript is well-structured and clear, but certain methodological and interpretational areas would benefit from further clarification and refinement.

Specific Comments:

Methods Section:

1. Use of the Moses Extreme Reaction Test:

The use of the Moses Extreme Reaction Test to compare group-level differences, such as between AIDS and non-AIDS stages, may not align with its intended application, which is primarily to detect extreme individual response patterns.

• Recommendation: Consider replacing this test with standard statistical methods for group comparisons. If the Moses Extreme Reaction Test is retained, please provide a detailed justification for its use in this context.

2. Clarity in Description of Measures:

The section describing the calculation of HRQoL metrics, including EQ-5D-3L and utility values, could be clarified and made more consistent for readers. The description in lines 161–185 is helpful but could be expanded for clarity.

• Recommendation: Provide a more uniform and detailed explanation of how each measure was calculated and interpreted to enhance readability and understanding.

Results and Discussion Sections:

1. Interpretation of “Areas for Improvement”:

The discussion of quality-of-life “areas for improvement” beginning on line 301 is not clearly linked to the presented results. While Table 3 outlines self-reported symptoms, it is unclear how these were identified as areas for improvement.

• Recommendation: Clearly describe the criteria or process used to identify these areas for improvement and ensure alignment with the results presented.

2. Duration of Follow-Up:

Conclusions about long-term health outcomes are based on data from a 48-week follow-up. While this provides initial insights, the extrapolation to longer periods may not fully reflect the available evidence.

• Recommendation: Acknowledge the limitation of the study’s follow-up period and note the need for longer-term studies to confirm these findings.

Strengths:

• The manuscript presents an interesting and valuable exploration of the pharmacoeconomic implications of BIC/FTC/TAF.

Minor Comments:

• Ensure all abbreviations, such as QALYs and HRQoL, are consistently defined at first use in the body of the manuscript.

Conclusion:

This manuscript provides relevant information regarding the clinical and economic implications of BIC/FTC/TAF for rapid initiation in naïve HIV-infected individuals. Addressing the comments outlined above will enhance clarity and rigor without requiring substantial changes.

Note: If required, it was not clear that all data underlying the findings in their manuscript will be made fully available to readers (authors did offer data for journal reviewers).

**Do you want your identity to be public for this peer review?** For information about this choice, including consent withdrawal, please see our Privacy Policy

Reviewer #1: No

Reviewer #2: No

---

## [Author Response · Author response to Decision Letter 1]

12 Feb 2025

Reviewer #1: General comments:

These analyses explore quality of life measures in a subset of clinical trial participants. Comparisons are made between participant scores 24- and 48-weeks post-recruitment.

-We have revised the tables and text to clarify that comparisons are generally made between values at baseline and at 48 weeks, only considering values at 24 weeks when they are of special interest for our study.

I have some concerns with these analytic methods. First, there is no attempt to control for confounders in any of the analyses. All differences are attributed to time (24 vs. 48 weeks post recruitment) and no other factors that may have changed are considered. A multivariable model that controls for other factors would be beneficial.

-Crucially, as now underscored in the revised manuscript, we studied the same individuals over the study period, with every patient serving as their own control. Interindividual variability was also reduced by our strict eligibility criteria, given that the study population exclusively comprised naïve patients started on BIC/FTC/TAF as a test-and-treat strategy. For these reasons, and given our study objective, we respectfully consider that the construction of a multivariable model is not essential.

A multivariable model would also help my second concern. I think the manuscript would benefit from having comparisons reported as differences with CIs. For instance, I found it hard to understand the difference in line 234 since, based on the SDs, it looks like the intervals might overlap. Reporting differences as mean differences or odds ratios with a CI would facilitate understanding how comparisons were made.

-As requested, we now report mean differences with CIs in the text (where significative p-values are given) and in a supplementary table.

Third, why is baseline not included in analyses? I would have expected that comparisons would be made to baseline whenever possible, yet the analyses and write up suggest baseline does not factor in to these analyses. I would love to see longitudinal analyses performed, probably with random effects, where confounders are included in the models and you could estimate 24w v. BL, 48w v. BL, and 48w v. 24w.

-As clarified in the text and tables, comparisons are made between baseline and week 48 except when otherwise indicated. We appreciate the reviewer’s suggestion of a random effect model, but we believe that this would add unnecessary complexity to the manuscript. Please also see our response to the first and second points above.

Third, I found the reporting of missing data and the sample sizes for analyses confusing. For instance, lines 215-216 have 208 people at 24-weeks and 178 at 48-weeks. That indicates to me that the two time points have different sample sizes. Thus, the differences quoted in lines 225-227 could be explained by participant drop out and not necessarily differences between 24- and 48-weeks. It's even worse in Tables 2 and 3 where the sample sizes are never 208. Differences in the sample size may not contribute to differences, but as currently presented, the differing sample sizes raise doubt on your results. I strongly recommend using the same sample size and participants for all analyses in a subset.

-As stated above, we always used the same patient database (all participants in the BICNOW clinical trial). However, as expressly documented, not all participants remained throughout the study, and some questionnaires were invalid. We analyzed all valid questionnaires available at each time point, and their number is reported in the tables. In this way, we believe that our study offersa robust and transparent analysis of real-world changes in the quality of life of these patients. 

In addition, if there are only 178 people, that means, based on your sample size, you are underpowered for your analyses. In addition, the sub-subgroup analyses of AIDS and non-AIDS people are further underpowered. Were there power analyses performed for these sub-subgroup analyses?

According to our estimation of the sample size (see supplementary information), it continues to offer adequate statistical power at week 48 (n= 178), when the loss to follow-up is similar to the percentage (10-15%) typically observed in real-life studies. We have added the following information on the sample size estimation for the sub-group of patients in AIDS stage:

"In the substudy of participants in AIDS stage, we estimated that a sample size of 40 patients would be sufficient to detect differences with 95% confidence interval and 85% power.”

Lastly, there are LOTS of evaluations being done here. I would think some sort of p-value correction should be performed to account for multiple testing. For instance, something controlling for the false discovery rate would be useful.

-We have addressed this concern with the following addition in Material and Methods, explaining why we did not apply a broad p-value correction:

"The questionnaires used to evaluate quality of life (EQ-5D, with 5 dimensions plus 1 VAS;and HIV-SI, with 20items) represent two validated measures rather than numerous independent assessment criteria. Each dimension or item is part of an interrelated set that captures specific aspects of quality of life or symptom burden. We therefore did not apply formal multiple comparison adjustments (e.g., Bonferroni or FDR), because they might unnecessarily inflate type II error rates for these intrinsically related dimensions. We focus on the main result (total score and/or key item) of each questionnaire, providing subdomain results to supply additional clinical information. All p-values should be interpreted in the context of these correlated measures and item-level analyses."

Specific comments:

1. (line 55) "p<" needs to be changed.

-This has been changed.

2. (lines 136-138) Are the symptoms dichotomized in the survey instrument? If they are not dichotomized in the instrument, then I believe this would invalidate the validation that was done on the instrument since the instrument is not used as developed.

-Yes, the symptoms are dichotomized, using the validated instrument as developed.

3. (lines 141-143) How many questionnaires were deemed invalid?

-At week 48, 9 EQ-5D-3L and 6 HIV-SI questionnaires were deemed invalid, as now noted in the revised text.

4. (line 195) Although the Kolmogorov–Smirnov test is frequently used, it is well-known to be underpowered for testing normality. Other tests have more power (e.g., see https://www.jstor.org/stable/2286009), such as Andersen-Darling and Cramer von Mises.

-We are grateful for this comment, which will be considered in future studies. On the other hand, this is the most frequently applied test in clinical research and is maintained in the present investigation.

5. (lines 195-202) I would prefer providing methodological citations for each of the tests here to credit test developers for their work, though it seems to be ok to not cite commonly used tests. I am familiar with all tests in this section, except for the Moses extreme test, so I would suggest providing a methodological citation for that test.

-We are grateful for this comment and now provide a methodological citation for the Moses extreme test.

Moses LE. Rank Tests of Dispersion. The Annals of Mathematical Statistics. 1963;34: 973–983. doi:10.1214/aoms/1177704020

6. (lines 219-227) I'm assuming the comparisons in this paragraph are between the 24- and 48-week outcomes, but it's not completely clear.

-As reported in table 2 and now clarified in the revised text, these comparisons are made between baseline and week 48.

7. (line 224) What is "profile 11111"?

-In accordance with the patient profiles generated by the EQ-5D-3L,"1" representsoptimal health (no problems) for each dimension, and "11111" therefore indicatesperfect health, as now explained in the same line.

8. Why are the tables out of order?

-The tables are now included in the text in the correct order.

9. (Figure 2) I don't think this figure adds anything that cannot be explained in the text.

-We would prefer to retain this reader-friendly depiction of the improvement in VAS results forthe EQ-5D-3L questionnaire improvement in self-perceived quality of life.

Reviewer #2: General Comments:

This manuscript addresses an important topic in HIV care by evaluating the impact of rapid initiation of bictegravir/emtricitabine/tenofovir alafenamide (BIC/FTC/TAF) on health-related quality of life (HRQoL) and economic outcomes. The findings contribute useful information regarding clinical and pharmacoeconomic aspects of this treatment strategy. The manuscript is well-structured and clear, but certain methodological and interpretational areas would benefit from further clarification and refinement.

Specific Comments:

Methods Section:

1. Use of the Moses Extreme Reaction Test:

The use of the Moses Extreme Reaction Test to compare group-level differences, such as between AIDS and non-AIDS stages, may not align with its intended application, which is primarily to detect extreme individual response patterns.

• Recommendation: Consider replacing this test with standard statistical methods for group comparisons. If the Moses Extreme Reaction Test is retained, please provide a detailed justification for its use in this context.

-We acknowledge that the Moses Extreme Reaction Test is not frequently applied for the comparison of means or medians. However, our intention was to explore between-group differences in variability or dispersion rather than central tendencies alone.

We suspect that individuals with AIDS may experience wider fluctuations or more extreme values in their quality-of-life measurements (e.g., EQ-5D, VAS) in comparison to those without AIDS. The Moses test is appropriate to detect score dispersion differences and the presence of atypical values/extreme responses.

Besides centering on extremes, the Moses Extreme Reaction Test is a non-parametric and therefore robust test when underlying distributions may be non-normal and when subgroup sample sizes are relatively small or unbalanced. This is especially relevant in the present study, in which the AIDS subgroup is smaller than the non-AIDS subgroup.

2. Clarity in Description of Measures:

The section describing the calculation of HRQoL metrics, including EQ-5D-3L and utility values, could be clarified and made more consistent for readers. The description in lines 161–185 is helpful but could be expanded for clarity.

• Recommendation: Provide a more uniform and detailed explanation of how each measure was calculated and interpreted to enhance readability and understanding.

-As recommended, we have extended the interpretation of definitions at the beginning of the section and have included relevant citations.

Results and Discussion Sections:

1. Interpretation of “Areas for Improvement”:

The discussion of quality-of-life “areas for improvement” beginning on line 301 is not clearly linked to the presented results. While Table 3 outlines self-reported symptoms, it is unclear how these were identified as areas for improvement.

• Recommendation: Clearly describe the criteria or process used to identify these areas for improvement and ensure alignment with the results presented.

-We now clarify this point in Material and Methods (136-8):

“Results for individual symptoms are used to identify areas for improvement in those that are most prevalent among participants.

2. Duration of Follow-Up:

Conclusions about long-term health outcomes are based on data from a 48-week follow-up. While this provides initial insights, the extrapolation to longer periods may not fully reflect the available evidence.

• Recommendation: Acknowledge the limitation of the study’s follow-up period and note the need for longer-term studies to confirm these findings.

-As recommended, we have added the following limitation:

Furthermore, the follow-up is restricted to 48 weeks, and longer-term research is needed.

Strengths:

• The manuscript presents an interesting and valuable exploration of the pharmacoeconomic implications of BIC/FTC/TAF.

-We are grateful for this appreciation.

Minor Comments:

• Ensure all abbreviations, such as QALYs and HRQoL, are consistently defined at first use in the body of the manuscript.

-This has been done.

Conclusion:

This manuscript provides relevant information regarding the clinical and economic implications of BIC/FTC/TAF for rapid initiation in naïve HIV-infected individuals. Addressing the comments outlined above will enhance clarity and rigor without requiring substantial changes.

Note: If required, it was not clear that all data underlying the findings in their manuscript will be made fully available to readers (authors did offer data for journal reviewers).

-It is now stated that all data underlying these findings are fully available to the readers.

---

## [Decision Letter · Decision Letter 1]

23 Mar 2025

Dear Dr. Sequera-Arquelladas,

Thank you for submitting your manuscript to PLOS ONE. After careful consideration, we feel that it has merit but does not fully meet PLOS ONE’s publication criteria as it currently stands. Therefore, we invite you to submit a revised version of the manuscript that addresses all the points raised during the review process.

If applicable, we recommend that you deposit your laboratory protocols in protocols.io to enhance the reproducibility of your results. Protocols.io assigns your protocol its own identifier (DOI) so that it can be cited independently in the future. For instructions see: https://journals.plos.org/plosone/s/submission-guidelines#loc-laboratory-protocols . Additionally, PLOS ONE offers an option for publishing peer-reviewed Lab Protocol articles, which describe protocols hosted on protocols.io. Read more information on sharing protocols at https://plos.org/protocols?utm_medium=editorial-email&utm_source=authorletters&utm_campaign=protocols.

We look forward to receiving your revised manuscript.

Kind regards,

Giuseppe Vittorio De Socio, MD, PhD

Academic Editor

PLOS ONE

Reviewers' comments:

Reviewer's Responses to Questions

**Comments to the Author**

Reviewer #1: (No Response)

2. Is the manuscript technically sound, and do the data support the conclusions?

Reviewer #1: Partly

3. Has the statistical analysis been performed appropriately and rigorously?

Reviewer #1: Yes

4. Have the authors made all data underlying the findings in their manuscript fully available?

Reviewer #1: Yes

5. Is the manuscript presented in an intelligible fashion and written in standard English?

Reviewer #1: Yes

Reviewer #1: Thank you for your careful consideration of my comments. There are still a couple comments where I continue to disagree.

First to this response:

"As stated above, we always used the same patient database (all participants in the BICNOW clinical trial). However, as expressly documented, not all participants remained throughout the study, and some questionnaires were invalid. We analyzed all valid questionnaires available at each time point, and their number is reported in the tables. In this way, we believe that our study offersa robust and transparent analysis of real-world changes in the quality of life of these patients."

In table 2, I presume your p-value is for the 169 participants who have week 48 observations. If so, what are the numbers that are going into that test? The week 48 value is, but the baseline is not because you have presented the value with n=186. That thinking suggests that the way you have presented table 2 is not transparent. Regarding your comment about robustness, I'm not sure I follow in a statistical sense since, presumably the 18 participants (186-169) don't contribute to the statistical analyses.

Second, regarding this response to my comment about Kolmogorv-Smirnov:

"We are grateful for this comment, which will be considered in future studies. On the other hand, this is the most frequently applied test in clinical research and is maintained in the present investigation."

If we as scientists followed this logic, we would never change our analytic methods. This doesn't appear to be a valid scientific reason, so I still encourage you to make this change.

**Do you want your identity to be public for this peer review?** For information about this choice, including consent withdrawal, please see our Privacy Policy

Reviewer #1: No

---

## [Author Response · Author response to Decision Letter 2]

28 Mar 2025

Reviewer #1: Thank you for your careful consideration of my comments. There are still a couple comments where I continue to disagree.

First to this response:

"As stated above, we always used the same patient database (all participants in the BICNOW clinical trial). However, as expressly documented, not all participants remained throughout the study, and some questionnaires were invalid. We analyzed all valid questionnaires available at each time point, and their number is reported in the tables. In this way, we believe that our study offersa robust and transparent analysis of real-world changes in the quality of life of these patients."

In table 2, I presume your p-value is for the 169 participants who have week 48 observations. If so, what are the numbers that are going into that test? The week 48 value is, but the baseline is not because you have presented the value with n=186. That thinking suggests that the way you have presented table 2 is not transparent. Regarding your comment about robustness, I'm not sure I follow in a statistical sense since, presumably the 18 participants (186-169) don't contribute to the statistical analyses.

RESPONSE-

We thank the reviewer for this comment. We have addressed this concern by incorporating data in revised Tables 2 and 3 on the number of observationsconsidered at each time point forthe p-value calculation.

Table 2. EQ-5D-3L – results per evaluated dimension

BASELINE N =186

24W N=178

48W N=169

p-value (BASELINE vs 48W) N=169

Table 3. Results of dichotomized HIV-SI questionnaire (bothersome/non-bothersome)

BASELINE N = 199

W24 N= 187

W48 N=172

p-value (BASELINE vs 48W) N=172

Second, regarding this response to my comment about Kolmogorv-Smirnov:

"We are grateful for this comment, which will be considered in future studies. On the other hand, this is the most frequently applied test in clinical research and is maintained in the present investigation."

If we as scientists followed this logic, we would never change our analytic methods. This doesn't appear to be a valid scientific reason, so I still encourage you to make this change.

RESPONSE-

As a result of this encouragement, we applied both tests proposed by the reviewer using version 4.4.3 of the R statistical package. There were no differences in normality results among the three tests (Kolmogorov-Smirnov,Cramer-von Mises,and Anderson-Darling tests). Following the reviewer’s suggestion, we selected the Cramer-von Mises test for the revised manuscript.

---

## [Editor Report · Decision Letter 2]

4 Apr 2025

Impact of Bictegravir/Emtricitabine/Tenofovir Alafenamide on health-related quality of life and economic outcomes in HIV care: substudy of the BIC-NOW clinical trial

PONE-D-24-49234R2

Dear Dr. Sequera-Arquelladas,

We’re pleased to inform you that your manuscript has been judged scientifically suitable for publication and will be formally accepted for publication once it meets all outstanding technical requirements.

Kind regards,

Giuseppe Vittorio De Socio, MD, PhD

Academic Editor

PLOS ONE
---

## [Editor Report · Acceptance letter]

PONE-D-24-49234R2

PLOS ONE

Dear Dr. Sequera-Arquelladas,

I'm pleased to inform you that your manuscript has been deemed suitable for publication in PLOS ONE. Congratulations! Your manuscript is now being handed over to our production team.

Kind regards,

on behalf of

Dr. Giuseppe Vittorio De Socio

Academic Editor

PLOS ONE